# A Review of Titanium Dioxide (TiO_2_)-Based Photocatalyst for Oilfield-Produced Water Treatment

**DOI:** 10.3390/membranes12030345

**Published:** 2022-03-19

**Authors:** Hadi Nugraha Cipta Dharma, Juhana Jaafar, Nurul Widiastuti, Hideto Matsuyama, Saied Rajabsadeh, Mohd Hafiz Dzarfan Othman, Mukhlis A Rahman, Nurul Natasha Mohammad Jafri, Nuor Sariyan Suhaimin, Atikah Mohd Nasir, Nur Hashimah Alias

**Affiliations:** 1Advanced Membrane Technology (AMTEC) Research Centre, School of Chemical and Energy Engineering, Faculty of Engineering, Universiti Teknologi Malaysia, Skudai 81310, Malaysia; nugraha-1999@graduate.utm.my (H.N.C.D.); hafiz@petroleum.utm.my (M.H.D.O.); mukhlis@petroleum.utm.my (M.A.R.); nnatasha6@live.utm.my (N.N.M.J.); sariyan@utm.my (N.S.S.); 2Department of Chemistry, Faculty of Science and Data Analytics, Institut Teknologi Sepuluh Nopember, Surabaya 60111, Indonesia; nurul_widiastuti@chem.its.ac.id; 3Center for Membrane and Film Technology, Graduate School of Engineering, Faculty of Engineering, Kobe University, Kobe 657-8501, Japan; matuyama@kobe-u.ac.jp (H.M.); rajabzadehk@people.kobe-u.ac.jp (S.R.); 4Centre for Diagnostic, Therapeutic & Investigative Studies (CODTIS), Faculty of Health Sciences, Universiti Kebangsaan Malaysia, Bangi 43600, Malaysia; atikahnasir@ukm.edu.my; 5Oil and Gas Engineering, School of Chemical Engineering, Universiti Teknologi MARA, Shah Alam 40450, Malaysia; nurhashimah@uitm.edu.my

**Keywords:** oilfield-produced water (OPW), photodegradation, photocatalyst, TiO_2_

## Abstract

Oilfield produced water (OPW) has become a primary environmental concern due to the high concentration of dissolved organic pollutants that lead to bioaccumulation with high toxicity, resistance to biodegradation, carcinogenicity, and the inhibition of reproduction, endocrine, and non-endocrine systems in aquatic biota. Photodegradation using photocatalysts has been considered as a promising technology to sustainably resolve OPW pollutants due to its benefits, including not requiring additional chemicals and producing a harmless compound as the result of pollutant photodegradation. Currently, titanium dioxide (TiO_2_) has gained great attention as a promising photocatalyst due to its beneficial properties among the other photocatalysts, such as excellent optical and electronic properties, high chemical stability, low cost, non-toxicity, and eco-friendliness. However, the photoactivity of TiO_2_ is still inhibited because it has a wide band gap and a low quantum field. Hence, the modification approaches for TiO_2_ can improve its properties in terms of the photocatalytic ability, which would likely boost the charge carrier transfer, prevent the recombination of electrons and holes, and enhance the visible light response. In this review, we provide an overview of several routes for modifying TiO_2_. The as-improved photocatalytic performance of the modified TiO_2_ with regard to OPW treatment is reviewed. The stability of modified TiO_2_ was also studied. The future perspective and challenges in developing the modification of TiO_2_-based photocatalysts are explained.

## 1. Introduction

Oilfield produced water (OPW) is the by-product of oil exploration and production, both onshore and offshore, and has become one of the most significant organic pollutants on a global scale [1,2]. The physicochemical properties of OPW are highly dependent on the geological age, depth, and geochemistry of the hydrocarbon-bearing formation, as well as the chemical composition of the oil phase in the reservoir and the process of adding chemicals during oil production [2]. OPW has been estimated to be over 80 million barrels per day on average throughout the world with an 80% water cut [3].

Furthermore, another study revealed that the chemical oxygen demand (COD) rate of OPW was 2250 ppm, containing a variety of organic chemicals, such as polycyclic aromatic hydrocarbons (PAHs), surfactants, benzene toluene ethylbenzene xylenes (BTEX), phenolic compounds, and organic acids [1,3,4]. Those organic compounds have an adverse effect on environmental and human health, such as high resistance to biodegradation, high mutagenic and carcinogenic risks, such as tumour and cancer, and high toxicity to marine living things [5]. Other biological impacts due to the presence of OPW include high risk in the endocrine, non-endocrine, and reproductive systems [6].

To prevent OPW contamination to the aquatic environment, conventional physical or biological processes have been applied, such as gravimetry and ultrasonic separation, coagulation/flocculation, adsorption, membrane bioreactor, biologically active filtration, and conventional activated sludge [7,8,9,10]. Although the physical processes exhibited dispersed oil separation from OPW, numerous drawbacks can be seen, such as high operation cost, long processing time, large space requirements, possibly producing secondary pollutants and the inability of pollutant degradation [7,8]. The biological process has been known to be the most cost-effective method but still faces some disadvantages, such as along-time process, more complex management, and easy contamination [9].

Currently, the photocatalysis process using semiconductors has been chosen to be favorable in degrading the organic pollutants in OPW as it does not require additional chemicals and generates harmless compounds, such as carbon dioxide (CO_2_), water (H_2_O), and mineral acids [9]. This process involves charge carrier transfer from the photo-excited semiconductor as shown in Figure 1, which can be summarized into two steps: photoexcitation of the semiconductor, which causes the separation of electrons (e^−^) and holes (h^+^) as charge carriers and the migration/recombination of charge carriers with the emission of heat or light [11,12]. Various semiconductors have been investigated for use as photocatalysts, such as titanium (IV) oxide (TiO_2_), zinc (II) oxide (ZnO), gallium arsenide, tungsten (VI) oxide (WO_3_), gallium phosphide, and cadmium sulfide [13].

Among the other semiconductors, TiO_2_ has become a promising material that has been widely used in photocatalytic processes due to its superior properties, such as excellent optical and electronic properties, high photocatalytic activity, high chemical stability, low cost, non-toxicity, and environmental friendliness [15,16]. Several reports have found the greater beneficial properties of TiO_2_ over the other semiconductors, including (1) a higher photodegradation rate of Procion Red MX-5B using TiO_2_ photocatalyst (0.34 h^−1^) rather than using zinc (II) oxide (ZnO, 0.25 h^−1^) and tin (IV) oxide (SnO_2_, 0 h^−1^) [17]; and (2) no Ti^4+^ ion was detected in the TiO_2_ slurries system after the photodegradation of phenol and even after the phenol was completely degraded. 

This phenomenon showcases the high stability and eco-friendliness of TiO_2_ compared to the ZnO suspension system, which contains a higher amount of Zn^2+^ (0.93 mM) [18,19]. In addition, the TiO_2_ photocatalyst reduced COD and PAHs rates in OPW by up to 87.1% and 50% under UV irradiation, respectively [20,21]. Table 1 shows several fundamental reactions in the photocatalytic degradation of organic compounds by TiO_2_ [22,23].

Currently, the crystalline phase of TiO_2_ has been classified into anatase, rutile, brookite, and TiO_2_(B) as shown in Figure 2 [24]. Those four crystal phases could be described as the octahedral structures of TiO_6_ that share their edges with the other octahedral structures. The anatase and rutile phases belong to the tetragonal type, but the brookite belongs to the orthorhombic type [25]. 

In the TiO_2_(B) phase, the Ti-O octahedral connection was similar to the anatase but with a different arrangement in the monoclinic crystal [26]. Rutile was the most thermodynamically stable, while anatase, brookite, and TiO_2_(B) were metastable [27,28]. Among those crystal phases, TiO_2_ anatase has long been known as the most photoactive due to a high density or localized state, slow charge carrier recombination, highly consequent surface-adsorbed radical hydroxyl, and high chemical stability [29,30].

Commonly, TiO_2_ can be synthesized using numerous routes, such as sol-gel, hydrothermal/solvothermal, sonochemical process, template method, electrodeposition, chemical and physical vapor deposition, aerosol synthesis, precipitation, and microelmusion-mediated, milling, sputtering, pulse laser ablation, and evaporation–condensation, which can be classified into two approaches: top-down and bottom-up [31,32,33,34,35,36]. Different precursors were used to obtain TiO_2_, such as titanium isopropoxide (TTIP), titanium tetrabutoxide (TBT), titanium tetrachloride (TiCl_4_), titanium alkoxide, and titanium trichloride (TiCl_3_), which could affect the particle size, morphology, and physicochemical properties [37,38,39]. 

Recently, a few studies have been conducted on the green synthesis of TiO_2_ utilizing plant-assisted and microbe-assisted approaches, which have advantages, such as non-toxicity, well-maintained reagent management, and a safe procedure [40]. Plant-assisted synthesis is more stable than microbe-assisted synthesis and has a higher yield but is less expensive [41,42]. Plant-assisted synthesis of TiO_2_ can be performed through several steps, including: plant extraction from any part (the flowers, seeds, leaves, and roots), mixing treatment of TiO_2_ precursor solution with plant extract, and post-treatment (drying and calcination) [40].

Although TiO_2_ has the potential to be one of the most efficient destructive photocatalysts for the majority of organic pollutants in OPW, it suffers from a low light absorption region and a low quantum yield, which hampers the formation of free radicals [43,44]. The capability of TiO_2_ is limited by the low rate of degradation for certain organic compounds, insufficient photon energy, and mass contact for TiO_2_ at the high concentration of organic pollutants [13]. Those limitations can occur because of the undesired TiO_2_ properties, such as low affinity toward organic pollutants and instability of nanosized particles, which may undergo aggregation and reduce the active sites [45]. Moreover, the application of the TiO_2_-suspended system for photodegradation of OPW faced major obstacles in terms of light scattering effects and complex photocatalyst recovery steps, which possibly reduced the photocatalytic performance [45,46,47].

To overcome those issues, several modifications of TiO_2_ were required to improve the photocatalyst performance in terms of boosting the charge carrier transfer, hindering the recombination of electrons and holes, and increasing the radiation light response up to the visible region. Numerous approaches have been established to achieve the improvement of TiO_2_ photocatalyst in the photodegradation process with comparative and beneficial properties, such as nanostructure modification, doping, and/or impregnation of other elements and/or material toward TiO_2_ photocatalysts and TiO_2_-based photocatalytic membranes (PMs) [45,48]. Hence, this review paper gives a detailed explanation regarding those approaches, various photodegradation mechanisms that have been found, and future perspectives for advancing TiO_2_-based photocatalysts in OPW treatment.

## 2. Modification Approaches of TiO_2_ Photocatalyst

### 2.1. Construction of TiO_2_ Nanostructures Form

The fabrication of TiO_2_ by modifying the morphological form hierarchically has been well studied and gained attention due to its diverse physicochemical properties, which can improve the photocatalytic performance [48]. This strategy has become one of the excellent routes to resolve challenges related to the kinetic and thermodynamic aspects of the TiO_2_ photocatalyst. Hierarchical nanostructure design for TiO_2_ has gained better molecular diffusion kinetics, enhanced radiation light absorption, increased surface area, and obtained excellent charge carrier separation with a lower possibility to recombination. 

Various nanostructure forms of TiO_2_ have been classified into 0D, 1D, 2D, and 3D architectures, which are summarized in Figure 3. Among those architectures, the TiO_2_ 1D has received the most attention and has been studied by the researchers due to its beneficial aspects, such as easy processing steps, large specific surface area, high aspect ratio and chemical stability, and efficient electronic charge properties, which can be synthesized using various techniques that are shown in Table 2 [48,49,50].

The development of other types of TiO_2_ 1D nanostructures has been attempted, and various types have been studied well. Interestingly, Xue et al. succeeded in constructing hierarchical TiO_2_ nanotrees, which were applied to the photodegradation of phenol under UV-light illumination [51]. TiO_2_ nanotrees are made up of TiO_2_ nanobelts or nanowires branches that are quasi-aligned vertically on a conductive substrate film, like a trunk, and can be made by immersing the trunk in alkali-hydrothermal and acidic conditions as shown in Figure 4. 

The scanning electron microscope (SEM) and atomic force microscope (AFM) images showed the nanosheet branches that were grown on the trunks with various lengths in the range of 50–200 nm, which follows the TiO_2_ nanotree illustration. The enhancement of photocatalytic activity for TiO_2_ nanotrees is significant, with the optimal condition exhibiting a photocatalytic reaction rate of 0.346 h^−1^, which is significantly higher than that of pristine TiO_2_ nanobelts (0.0256 h^−1^) and TiO_2_ nanowires (0.0154 h^−1^), despite the fact that the trunk thickness had no significant effect on the photoactivity of TiO_2_ nanotrees. This study has revealed potential avenues for scaling up TiO_2_ nanostructure modification for organic compound photodegradation in OPW.

Although TiO_2_ has been known for its beneficial effect on photocatalytic activity, there are some gaps that still exist in the implementation. There is no major difference, especially regarding the affinity property of TiO_2_ toward organic compounds in OPW because this strategy only changes the stability of crystal phases and morphological form and not the molecular properties of TiO_2_, such as intermolecular bonding to other compounds. Moreover, the use of TiO_2_-nanostructure modified still relies on the photocatalytic suspended system, which has limited performance due to the light scattering effect and higher pollutant concentration [52].

### 2.2. Energy Band Engineering for TiO_2_

The pristine TiO_2_ has a larger band gap, which only absorbs UV light for charge carrier separation and photoactivation, which utilizes 4% of the energy in the solar spectrum while 45% of the energy belongs to visible light [53]. 

Therefore, the energy band engineering strategy plays an important role in enhancing the longwave-light sensitivity of the TiO_2_ photocatalyst with excellent photocatalytic performance by narrowing and expanding the band gap of TiO_2_ into the wide region of light absorption using different approaches as illustrated in Figure 5. This strategy can be performed through various methods, such as developing composites or mixed-phase and doping them with suitable materials that have the heterojunction system, desired morphological, and physicochemical properties that result in minimizing the charge carrier recombination and optimizing photoexcited charge carriers between their interfaces [54,55,56].

The doping of TiO_2_ was conducted by introducing impurity ions, both non-metal and metal elements, as dopants into the TiO_2_ lattice plane without changing the single phase of TiO_2_ to substitute the host cations and/or anions, which could give a contribution in three routes: (1) increasing the conductivity of TiO_2_ and the mobility of photoexcited charge carriers that lead to the reduction of photogenerated electron/hole recombination and enhancement of charge carrier separation; (2) advancing the enlargement of the light absorption region and narrowing the band gap; and (3) altering the CBM of TiO_2_, which may enhance the optical properties [56,57,58,59,60]. 

Those contributions led to the appearance of a modified photocatalytic mechanism of TiO_2_ as shown in Figure 6, which can be further described as followed: (a) the non-metal doping on a TiO_2_ photocatalyst gained in impurity energy level, narrowed band gap, and improved oxygen vacancies, which is favorable to extend the visible-light absorption and prevent reoxidation [61,62,63,64]; and (b) the metal doping of TiO_2_ generated a new energy level in the band gap of TiO_2_, which may increase the amount of charge carrier trapping site and reduce the possibility of photogenerated electron/hole recombination [65].

Various experiments have been conducted by researchers to develop the doping of TiO_2_ for treating organic pollutants that were possibly presented in OPW. Wang et al. synthesized the tungsten (W)-doped TiO_2_ nanoparticles synthesis using the sol-gel method for reducing the viscosity of oilfield sewage under the mercury lamp illumination [66]. The W-doped TiO_2_ exhibited better photoactivity than undoped TiO_2_ with the optimum conditions at 1% of W-doped TiO_2_ loading, pH of 3 in the sol-gel process, and 3% of W loading on TiO_2_.

Other dopants for TiO_2_ were also applied for advancing the photocatalytic process toward organic compounds that are presented in OPW, which is summarized in Table 3. The success of the photodegradation process using the doped TiO_2_ can be determined by the appearance of an intermediate as the result of organic pollutant oxidation by reactive oxygen species (ROS), such as the hydroxyl radical (•OH) as illustrated in the example for the degradation mechanism of 4-nitrophenol (4-NP) by B-doped TiO_2_ in Figure 7 [67].

According to Dahl et al., the synthesis of nanoscale TiO_2_ composite material has been studied and widely reported in approximately 11,000 research articles that indicated the feasibility of this strategy for advancing TiO_2_ photocatalyst [74]. The TiO_2_-basedcomposite consists of more than one crystal phase in the solid phase, such as mixed-phase TiO_2_ or composite with non-TiO_2_ material.

The TiO_2_ composite could utilize wide range of light absorption up to visible light through a heterojunction effect, which conducted better photocatalytic performance by increasing the separation efficiency of photoexcited charge carriers between their interface or enhancing the adsorption capacity using the support material as depicted in Figure 8 [54,55,75]. Numerous TiO_2_-based heterostructured photocatalyst have been synthesized and applied for degrading numerous organic compounds that possibly appeared in OPW with improved performance and gained appropriate surface area and band gap as shown in Table 4.

Moreover, several studies showed that there was an improvement in the performance of mixed-phase TiO_2_, such as anatase-rutile (AR) and anatase-TiO_2_(B) compared to a single phase of TiO_2_ [76,77,78,79]. In addition, Ai et al. reported that TiO_2_ intercalated talc nanocomposites photodegraded more than 99.5% of 2,4-dichlorophenol (2,4-DCP) in 1 h under high-pressure mercury UV-lamp illumination (λ = 365 nm) while able to maintain high stability after 20 cycles (99% of 2,4-DCP degradation). This was due to the strong absorption ability of talc, which improved the power of capture–recombination carriers and photon absorption [80].

Numerous TiO_2_-based composite materials have shown their excellent properties, including the capability to tune the surface properties, such as open coordination sites, acidity/basicity, and hydrophilicity, which play important roles in the adsorption process as one of the critical factors relevant to photocatalysis and further modification [74]. Chekem et al. synthesized the activated carbon (AC)-TiO_2_ composite material by coating titania on the AC surface, which has been applied in the adsorption/photodegradation of phenol as illustrated in Figure 9 [87].

Research reported that AC-TiO_2_ with 25 wt% of TiO_2_ (CAT25) showed a higher adsorption capacity due to a larger surface area (q = 124.5 mg/g, S_BET_ = 609 m^2^/g) compared with pristine AC material (q = 117.4 mg/g, S_BET_ = 571 m^2^/g), which led to the enhancement of the adsorption/photodegradation process until approximately 95% of phenol degradation has been achieved under UV light illumination (λ = 270 nm). In addition, another study revealed the feasibility of Fe^3+^-TiO_2_-Zeolite composite material in advancing the diminution of COD concentration in oilfield wastewater both under sunlight and mercury UV-lamp irradiation by reducing the lower band from 3.06 eV (405 nm) to 2.95 eV (420 nm) [88].

Furthermore, the improvement of Fe^3+^-TiO_2_-Zeolite composite performance in the photocatalytic process was due to the involvement of Fe^3+^ in trapping electrons at the composite surface and transferring electrons to the adsorbed oxygen molecule as shown in Table 5. Syed et al. also reported that the epoxy resin/nano-TiO_2_ composite material as a photocatalyst showed a higher removal of organic pollutants in OPW under direct sunlight radiation in terms of reducing dissolved oxygen (DO), COD, turbidity, conductivity, and total dissolved solids [89].

The TiO_2_-based composite photocatalyst has been utilized further for treating saline-produced water (SWP), which is similar to OPW but mixed with saline water. The photocatalytic performance of the pristine TiO_2_ failed to degrade organic pollutants in SWP due to the high concentration of chloride ion (Cl^−^) that can neutralize HO^•^ species and inhibit the chemical bond breaking of organic species [90].

Andreozzi et al. invented a graphene-like TiO_2_ nanocomposite (rGO-TiO_2_) using the facile hydrothermal method with high desired physicochemical properties that applied band-energy engineering, such as high surface area (up to 179.9 m^2^/g), uniform distribution of mesopore, and a slightly lower band gap compared to the pristine TiO_2_ because of the formation of intraband gaps upon the incorporation of rGO [91]. Furthermore, rGO-TiO_2_ with 10% amount of rGO achieved the greatest reduction of total organic carbon (TOC) among the other weight ratios (1%, 5%, and 20%) and bare TiO_2_ as in Figure 10A and demonstrated the ability to degrade common organic species in OPW, such as acetic acid, phenol, naphthalene, xylene, and toluene as shown in Figure 10B [91].

The improvement of TiO_2_-based photocatalyst can be gained by combining nanostructure modification and energy band engineering, which is expected to have much higher performance in degrading organic species in OPW. Rongan et al. managed to fabricate S-scheme photocatalyst Bi_2_O_3_/TiO_2_ nanofibers (NFs) via electrospinning and in-situ photo reductive deposition processes [92].

It was confirmed that Bi_2_O_3_/TiO_2_ NFs possessed well desired physicochemical properties, such as high surface area (51 m^2^/g), uniform mesoporous, red-shift light absorption, and hierarchical microspheres morphology with the size around 0.5–1 µm. Moreover, Bi_2_O_3_/TiO_2_ NFs showed better performance than the bare Bi_2_O_3_ and pristine TiO_2_ in phenol photodegradation due to the unique hierarchical porous structure as illustrated in Figure 11.

Another combination of energy band engineering and nanostructure modification for advancing photoactivity of TiO_2_ was revealed by Jo et al. [93]. In their study, hydrothermal process has been applied to construct TiO_2_ nanotubes-coated carbon fibers (TNTUCF) with various precursor concentrations, such as 5%, 7.5%, 10%, 12.5%, and 15%. From BTEX photodegradation test under UV illumination as in Figure 12, it has been shown that TNTUCF at 10% concentration of the precursor gained the highest degradation efficiencies of BTEX at 81%, 97%, 99%, and 99%, respectively.

Moreover, the photoactivity of TNTUCF has been significantly enhanced compared to TiO_2_ nanoparticle-coated CF (TUCF) and uncoated CF (UCF). The combined effect between large adsorption capacity and reduced recombination rates of electrons and holes on TNT explained the high performance of TNTUCF toward photocatalytic degradation of BTEX. However, excess amount of precursor concentration caused coverage of active sites on CF surface that may decrease the adsorption capacity of TNTUCF.

Wang et al. invented TiO_2_ photoactivity enhancement by producing a heterostructured palladium oxide-loaded corroded TiO_2_ nanobelts (PdO-C-NB) through two stages, including: hydrothermal and chemical precipitation processes [94]. This advanced material integrated beneficial aspects of TiO_2_ nanobelts, such as large surface area, feasible synthesis methodology, and controllable morphology, as well as heterojunction that could increase the separation of photo-induced electron-hole pairs and encourage redox reaction.

Furthermore, corroding nanobelts was a good approach for roughening the surface, which boosts surface active sites and specific surface area, providing a more appropriate surface for heterogeneous nucleation. The increased specific surface area and decreased recombination of photoexited charge carriers were ascribed to the enhanced photocatalytic activity of PdO-C-NB, as demonstrated by N_2_ adsorption-desorption and PL tests, as shown in Table 6 and Figure 13a.

As an outcome in Figure 13b, PdO-C-NB exhibited higher phenol degradation up to 61% within 90 min under UV light, compared to TiO_2_ nanobelts (NB), which acquired only 35%. Moreover, a photocatalytic mechanism was proposed, as in Figure 13b, in which the separation of electron-hole pairs was further driven by transferring electrons from TiO_2_ nanobelts to PdO, which could hinder recombination of those pairs and optimize photodegradation performance.

### 2.3. TiO_2_-Based Photocatalytic Membrane (PM)

Performing photocatalysis process standalone using the TiO_2_-based semiconductor toward OPW pollutant might not be effective for a longer time due to numerous factors that can interfere with the effectiveness and stability of photodegradation performance like complex composition in OPW, light scattering effect due to high concentration of organic species, and closure of active site on photocatalyst surface, which exacerbates the mass transfer process [95]. Therefore, the incorporation of photocatalyst with other suitable technology seems to be a good option for addressing issues related to OPW treatment on a large scale. This combination can be conducted by taking into account the various changes that have occurred, such as rate of OPW degradation, physicochemical properties of treated OPW, and endurance for long-term continuous process.

The PM became familiar with advanced technology that unites the membrane separation process with photodegradation of photocatalyst. In general, the PM produced better performance in terms of reducing membrane fouling, minimizing light scattering effect, further degrading of organic species, and improving permeate flux, and quality under light irradiation [52]. Due to those beneficial aspects, the PM also can be applied in OPW treatment. The treatment process using PM can be summarized into two types—that is, non-pressure and pressure-driven as shown in Figure 14. Several researchers have put great effort to incorporate TiO_2_-based photocatalyst on membrane with different modules, and the details are displayed in Table 7.

Polymeric matrix membranes are commonly used as support media for TiO_2_-based material attachment in the construction of TiO_2_-based PM. However, this fabrication generates a large amount of waste, which causes the TiO_2_-based PM to not be environmentally friendly. Worse, the polymeric structure of PM can be degraded under exposure to light radiation. Labuto et al. investigated the detachment of numerous polymeric membranes into aqueous solution, which has been proven through the determination of the nanoparticle concentration presented during exposure of UV-light as shown in Table 8 [101].

To promote green synthesis and sustainability, free-standing TiO_2_-based PM has been employed without using polymeric membrane supports. Liu et al. synthesized free-standing Ag/TiO_2_ nanofiber membranes by electrospinning, followed by dispersion and vacuum filtration, which demonstrated improved photodegradation performance in terms of degradation and constant rate up to 80% and 0.0211 min^−1^, respectively, as well as higher permeate flux compared to a P25-deposited membrane and TiO_2_ nanofiber membrane [102].

Song et al. constructed self-standing photocatalytic membranes from Zr-doped TiO_2_ nanofibrous via electrospinning, followed by a calcination process, that had superior tensile strength up to 1.32 MPa due to a well-connected and continuous netlike structure that maintains structural integrity even when bent, thus, demonstrating their superior structural flexibility [103]. Shen et al. incorporated TiO_2_ on C_3_N_4_-decorated carbon-fiber (CF) cloth as a filter-membrane-shaped photocatalyst to enhance the separation efficiency and life-time of photogenerated carriers via heterojunction that boosted the photocatalytic degradation performance up to 87% after seven grades, which was higher than CF/C_3_N_4_ (60%) and CF/TiO_2_ cloth (8%) as presented in Figure 15 [104].

## 3. Challenges and Future Development

TiO_2_-based photocatalyst technologies are promising for OPW treatment due to high photoactivity with excellent and high desired physicochemical properties, high chemical stability, low toxicity, and environmental friendliness. Those superiorities can also be applied to treating other organic pollutant sources, such as textile dye, organic solvent, petroleum hydrocarbons, detergent, and plastics [13,105]. Despite the fact that extensive and comprehensive investigations of advanced TiO_2_ modification have been conducted, until combinations of the approaches above in terms of free-standing TiO_2_-based PM can be studied and applied, numerous aspects of the photodegradation process have yet to be thoroughly elucidated. These include:(1)There is not much research focused on treating OPW directly, as well as optimization of photodegradation performance, taking into account numerous aspects, such as pH condition, pollutant concentration, organic species variety, and light source and intensity, which affect the quality and sustainability of those technologies themselves. Apart from the environmental consequences of high organic effluent concentration in OPW, the amount of energy required has a significant impact on the photocatalysis process. This is one of the reasons why solar photocatalysis has recieved so much interest recently.(2)A life cycle assessment (LCA) of those technologies that are needed for large scale application, especially the treatment of OPW. LCA is one of the most important tools for identifying the environmental effects of a process, as well as its feasibility and costs. Although numerous studies have applied for photodegradation, there is still a lack of understanding about their end of life.(3)In terms of industrial scale feasibility, whether the photocatalytic process should be employed as a pre-treatment or as an independent treatment for OPW degradation become major constraints.

Therefore, efforts should be made to enhance sustainability, which is expected to achive high continuous performance under solar radiation over a wider range of working conditions. To achieve this, numerous developments can be proposed as follows:Determining the most appropriate semiconductor and/or doping element that boosts TiO_2_ photocatalytic performance and stability, which is in agreement with physicochemical properties, such as the active surface sites, morphology, optical, and electron transfer properties. As a result, both teoritical and experimental aspects can be supported by this approach.Applying the photodegradation process of the OPW model solution using common organic pollutant species, such as phenol, BTEX, etc. Then, more in-depth thermodynamic and kinetic analyses are also possible.Conceptualizing the process flow design of OPW photodegradation process using high novelty TiO_2_-based PM. This needs to be considered in terms of techno-economic and LCA parameters so that the sustainability of TiO_2_-based PM can be optimized for long-term industrial applications.

## 4. Conclusions

This review summarized the TiO_2_-based photocatalyst as a game-changing technology that can be improved to address limitation of TiO_2_ semiconductors for OPW treatment. Numerous approaches have been explained in-depth in this review, such as energy band engineering, the construction of TiO_2_ nanostructures, and the development of TiO_2_-based PM. The success of these strategies can be determined by observing their morphological form, optical absorption, photocatalytic performance, and sustainability.

Moreover, uniting photocatalysts with other technologies could create a sustainable and stable treatment process system, such as free-standing TiO_2_-based PM, which is a combination of photocatalysis capability and a membrane separation process. LCA, research, and the development of TiO_2_-based PM are expected to be economically viable with the ability to solve current challenges and bring application to the industrial scale.

## Figures and Tables

**Figure 1 membranes-12-00345-f001:**
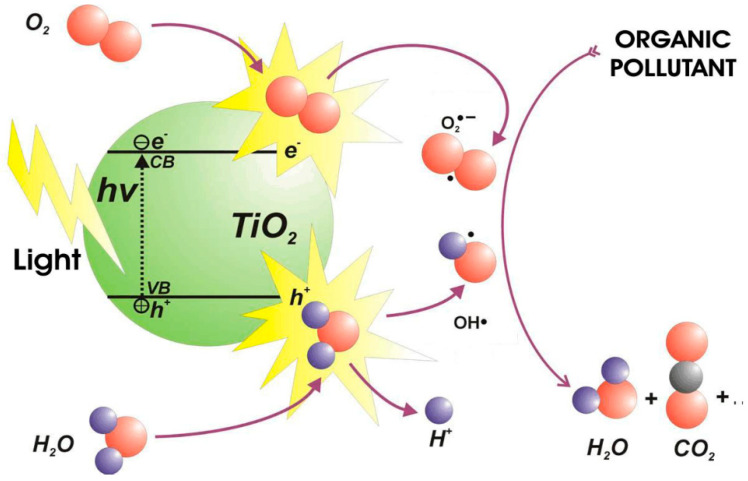
Basic principles of photodegradation process using semiconductors (example: TiO_2_) [14].

**Figure 2 membranes-12-00345-f002:**
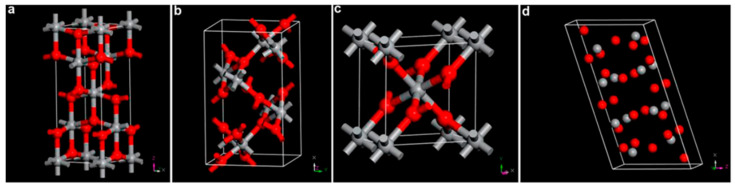
Representative crystal structure form of TiO_2_: (**a**) rutile, (**b**) brookite, (**c**) anatase, and (**d**) TiO_2_(B). Grey and red spheres represent the Ti^4+^ and O^2−^ ions, respectively [24].

**Figure 3 membranes-12-00345-f003:**
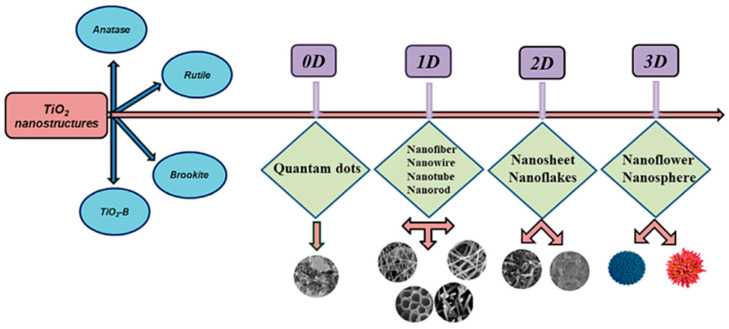
Categorization of hierarchical TiO_2_ nanostructure form [48]. Reproduced with permission from Elsevier, 2022.

**Figure 4 membranes-12-00345-f004:**
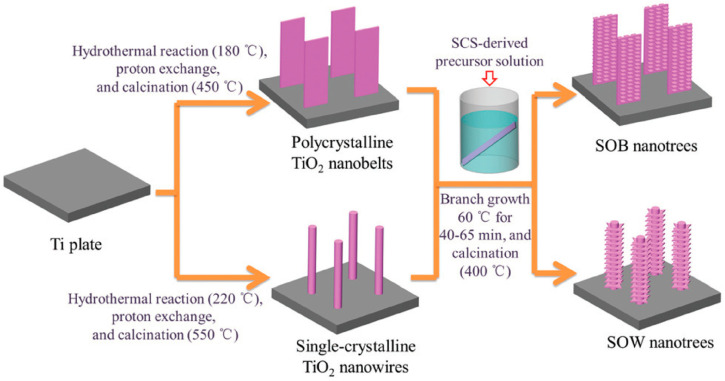
Synthesis steps of TiO_2_ nanotrees that can be divided into sheeton-belt (SOB) and sheet-on-wire (SOW) [51]. Reproduced with permission from the Royal Society of Chemistry, 2022.

**Figure 5 membranes-12-00345-f005:**
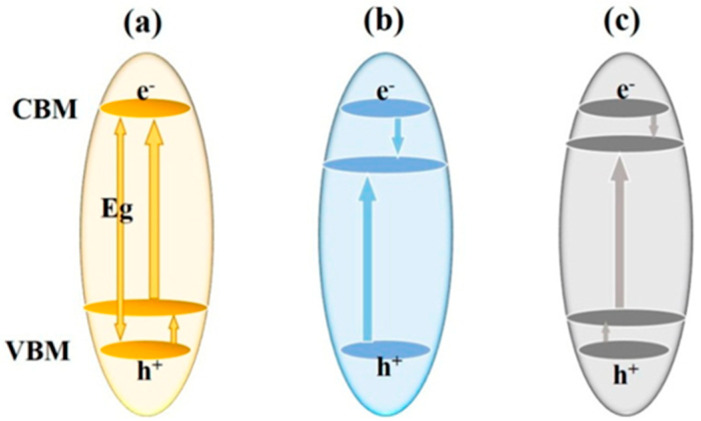
Representative mechanisms of the energy band engineering of TiO_2_: (**a**) a higher shift in valance band maximum (VBM); (**b**) a lower shift in conduction band minimum (CBM); and (**c**) continuous modification of both VBM and CBM [56].

**Figure 6 membranes-12-00345-f006:**
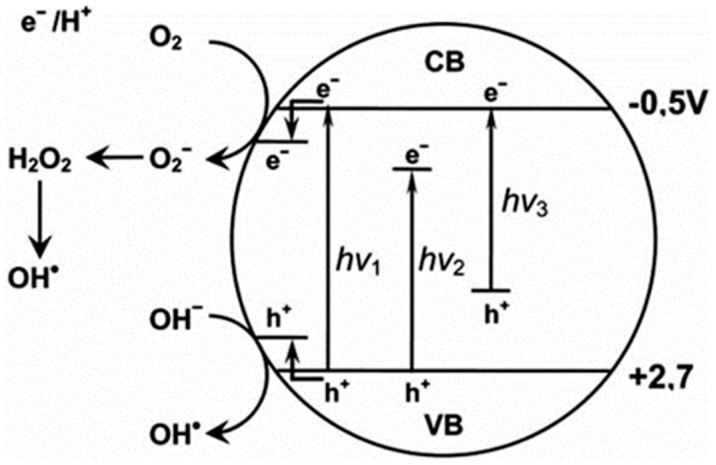
Representative scheme of photocatalysis process of TiO_2_. hν_1_: pristine TiO_2_; hν_2_: metaldoped TiO_2_; and hν_3_: nonmetaldoped TiO_2_ [65]. Reproduced with permission from Eureka Science (FZC), 2022.

**Figure 7 membranes-12-00345-f007:**
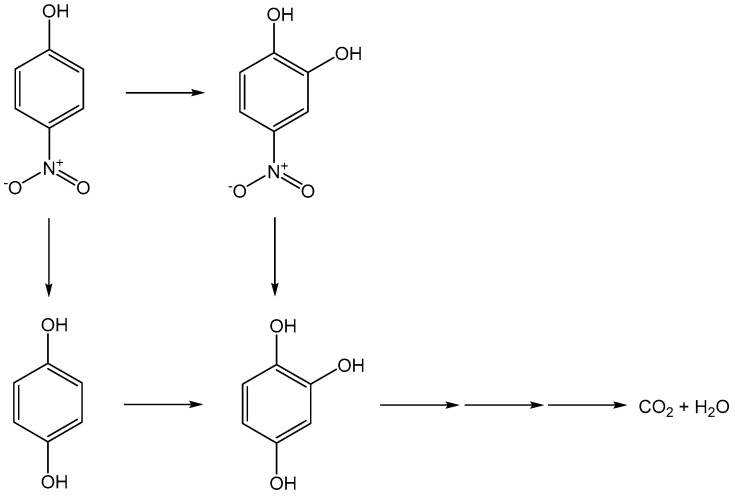
Route of 4-NP degradation by B-doped TiO_2_ under UV irradiation, adapted from [67].

**Figure 8 membranes-12-00345-f008:**
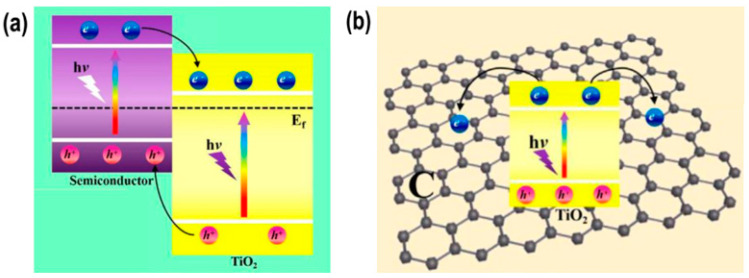
(**a**) Charge carrier migration mechanism of TiO_2_-based nanocomposite material, including trapping on TiO_2_ and the second phase/component and (**b**) TiO_2_ intercalation on the support material (example: graphene), which increases the light absorption region and intensity, charge carrier life-time, and absorptivity toward organic pollutants [81].

**Figure 9 membranes-12-00345-f009:**
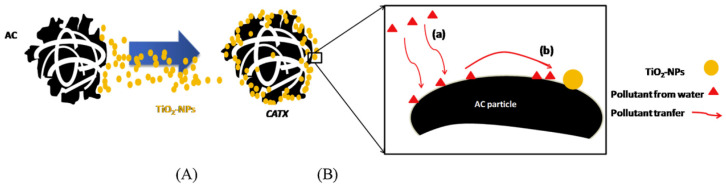
Scheme of (**A**) TiO_2_ impregnation at the external porosity of AC support by surface coating (CATX, X referred to %wt of TiO_2_) and (**B**) pollutant transfer towards photocatalytic center [87]. Reproduced with permission from Elsevier, 2022.

**Figure 10 membranes-12-00345-f010:**
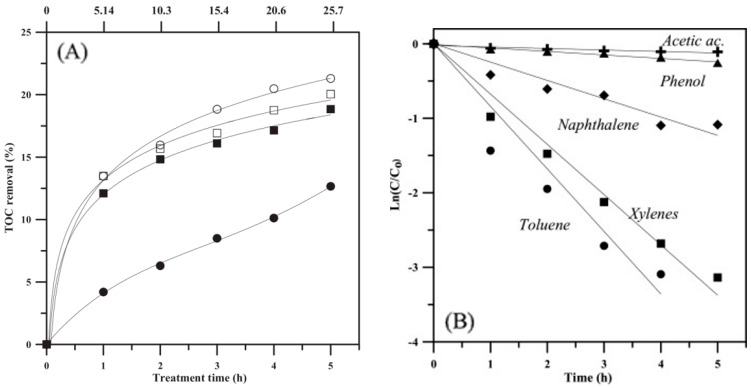
(**A**) TOC removal degree of bare TiO_2_-P25 (●), bare TiO_2_-HM (⬛), rGO(10%)/TiO_2_-P25 (○); rGO(10%)/TiO_2_-HM (□) with catalyst load at 500 mg/L and (**B**) linear plot of TOC degradation with various organic species in SPW [91]. Reproduced with permission from Elsevier, 2022.

**Figure 11 membranes-12-00345-f011:**
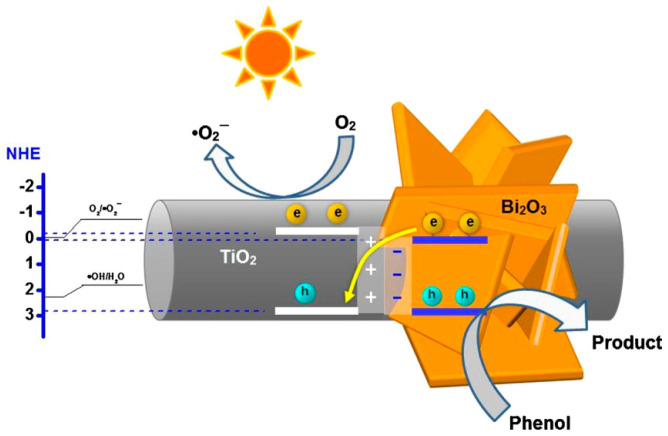
Schematic mechanism of phenol removal by photocatalysis using Bi_2_O_3_/TiO_2_ NFs [92]. Reproduced with permission from Elsevier, 2022.

**Figure 12 membranes-12-00345-f012:**
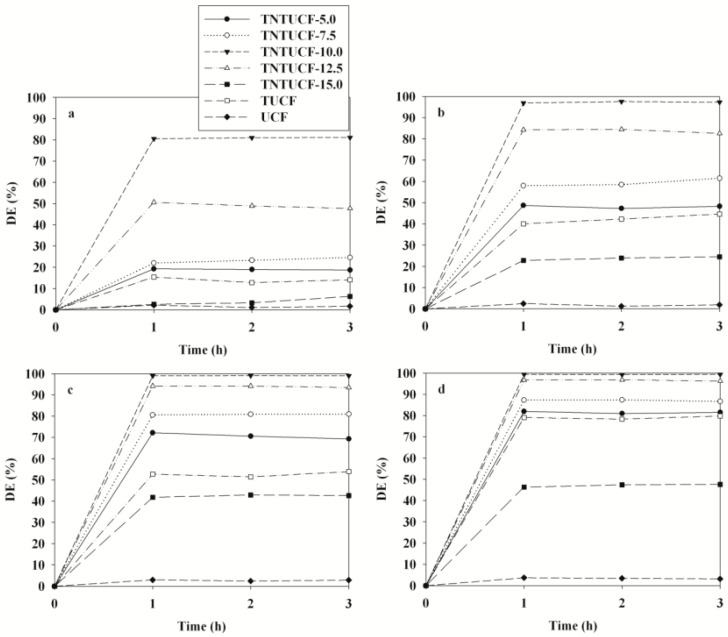
Comparison of degradation efficiencies (DE) of (**a**) benzene, (**b****)** toluene, (**c**) ethylbenzene, and (**d**) o-xylene under UV light irradiation by TNTUCF, TUCF, and CF [93].

**Figure 13 membranes-12-00345-f013:**
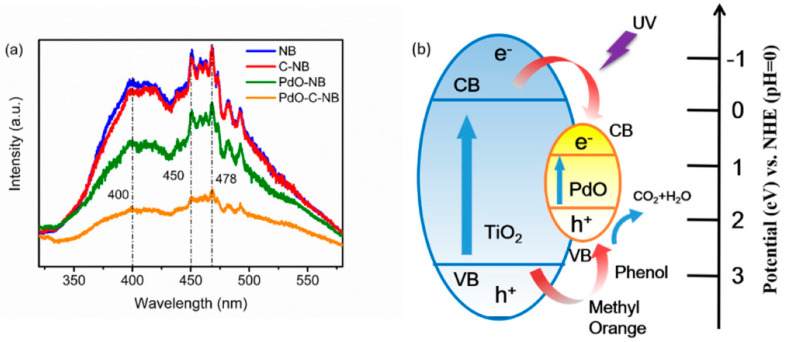
(**a**) PL spectra of photocatalyst samples and (**b**) schematic illustration of energy-band matching in PdO-C-NB for photodegradation of phenol under UV light irradiation [94]. Reproduced with permission from Elsevier, 2022.

**Figure 14 membranes-12-00345-f014:**
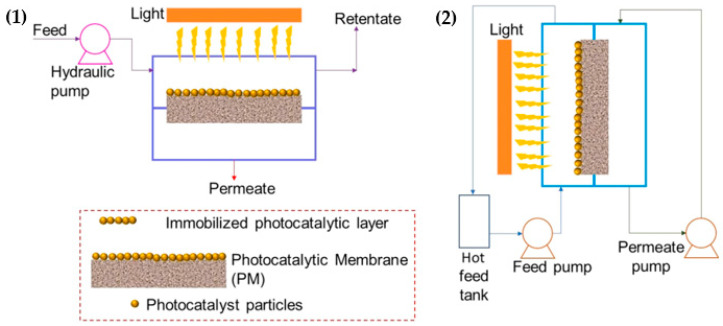
Schematic diagram of PM: (**1**) pressure-driven and (**2**) non-pressure driven [96]. Reproduced with permission from Elsevier, 2022.

**Figure 15 membranes-12-00345-f015:**
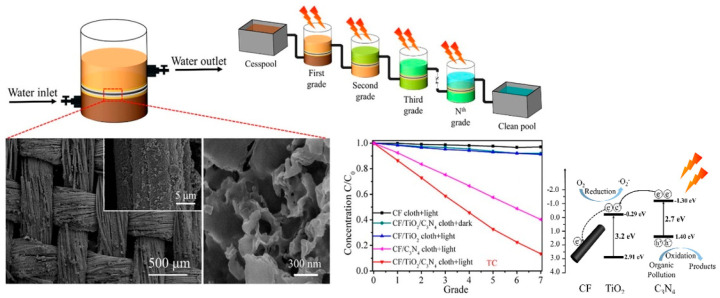
Summary of the PM process using a TiO_2_/C_3_N_4_/CF cloth filter membrane for pollutant degradation under light radiation with high desired morphological, good performance, and schematic mechanism of electron-hole separation [104]. Reproduced with permission from Elsevier, 2022.

**Table 1 membranes-12-00345-t001:** Various basic reactions with the respective reaction times for the general process of organic pollutant degradation by TiO_2_ photocatalyst [22,23]. Adapted with permission from American Chemical Society and Elsevier, 2022.

No.	Fundamental Reaction		Reaction Time (s)
1.	Generation of charge carriers	(1)	~10^−15^ (very fast)
	$TiO_{2}+h\nu \;\left(light\right)\to e_{cb}^{-}+h_{vb}^{+}$		
2.	Trapping charge carriers	(2)	10^−8^ (fast)
	$h_{vb}^{+}+Ti^{IV}OH\to {\left\{Ti^{IV}OH^{•}\right\}}^{+}$		
	$e_{cb}^{-}+Ti^{IV}OH\to \left\{Ti^{III}OH\right\}$	(3)	10^−10^ (a)
	$e_{cb}^{-}+Ti^{IV}\to Ti^{III}$	(4)	10^−8^ (b)
3.	Charge carrier recombination	(5)	10^−7^ (slow)
	$e_{cb}^{-}+{\left\{Ti^{IV}OH^{•}\right\}}^{+}\to Ti^{IV}OH$		
	$h_{vb}^{+}+\left\{Ti^{III}OH\right\}\to Ti^{IV}OH$	(6)	10^−8^ (fast)
4.	Interface charge carrier	(7)	10^−7^ (slow)
	${\left\{Ti^{IV}OH^{•}\right\}}^{+}+organic\;pollutant\to Ti^{IV}OH+oxidized\;pollutant$		
	$\left\{Ti^{III}OH\right\}+O_{2}\to Ti^{IV}OH+O_{2}^{•-}$	(8)	~10^−6^ (very slow)

Annotation: (a) shallow trap, dynamic equilibrium; (b) deep trap, irreversible; {Ti^IV^OH^•^}^+^ = result of hole trapping in the valence band on the surface of Ti(IV); and {Ti^III^OH} = result of electron trapping in the conductance band on the surface Ti(III).

**Table 2 membranes-12-00345-t002:** Several synthesis strategies for obtaining 1D TiO_2_ photocatalysts [48,50].

1D TiO_2_ Nanostructures	Synthesis Routes
TiO_2_ nanorods	sol-gel template, chemical vapor deposition (CVD), hydrothermal, wet-chemical
TiO_2_ nanotubes	template-assisted, hydrothermal, electrochemical deposition, sol-gel
TiO_2_ nanowires	hydrothermal, microwave-assisted, sol-gel electrophoretic deposition, solvothermal
TiO_2_ nanofibers	hydrothermal, electrospinning + sol-gel, hydrolysis
TiO_2_ nanobelts	solvothermal, hydrothermal, CVD

**Table 3 membranes-12-00345-t003:** Various preparations, applications, and results of the doped TiO_2_.

Doped Element(s)	Application	Method	Band Gap (eV)	Result		Reference
Boron (B)	4-NP degradation	Sol-gel	2.95–2.98	Under UV-light illumination, B-TiO_2_ generated 90% degradation of 4-NP (k = 0.0322 min^−1^), which was 11% higher than the pure TiO_2_ (k = 0.006 min^−1^). This improvement can be explained by the formation of Ti^3+^ due to B doping, which was responsible for enhancing the entrapment of photogenerated electron/hole as per the below reaction:		[67]
				$B+3Ti^{4+}\to B^{3+}+3Ti^{3+}$	(9)	
Silver (Ag)	Photodegradation of nitrophenol	Sol-gel involving reducing agent	3.1	Optimized Ag-TiO_2_ anatase with the size of 6 nm, photodegradation reaction rate at 0.034 min^−1^, which was almost 1.9-times higher than the pure TiO_2_.		[68]
Boron (B)	Phenol degradation	Sol-gel or Grinding- annealing	2.85	The B doping on TiO_2_ hindered the phase transformation from amorphous to anatase but can be active in the presence of visible light. However, excessive addition of B dopant resulted in B_2_O_3_ formation, which hampered the photoactivity.		[69]
Ag-Sulfur (S)	Photocatalytic removal of 2-nitrophenol (2-NP)	Sol-gel	2.39	Under the solar experiment, the photodegradation and adsorption rate constant of Ag-S/TiO_2_ (Eg = 2.39 eV) obtained were 2.4- and 4.1-times larger than the bare TiO_2_ (Eg = 3.05 eV), respectively. The BET surface area and pore volume of Ag-S/TiO_2_ were found to be higher up to 60.6% and 30%, respectively.		[70]
Transition metal (Iron (Fe), Vanadium (V), and Tungsten (W))	Photocatalytic degradation of BTEX	Solvothermal	3.14 (Fe-TiO_2_); 2.9 (W-TiO_2_ and V-TiO_2_)	The visible light catalytic activity of all doped TiO_2_ under is greater than that of pure TiO_2_. The V-TiO_2_ shows the highest photocatalytic activity followed by the W-TiO_2_ (54% conversion) and Fe-TiO_2_ (48% conversion) with the conversion of 69%. The UV-Vis diffuse reflectance spectra reveal that the V-TiO_2_ has the highest visible light absorption followed by the W-TiO_2_, Fe-TiO_2_, and undoped TiO_2_. The enhancement of TiO_2_ by transition metal doping is most likely due to the effect of the highest increase in visible light absorption as well as the smallest particle size among the V- TiO_2_ doped samples.		[71]
Mangan (Mn)	Photooxidation of benzene, toluene, and xylene	Sol-gel	3	Under UV irradiation and in the appearance of oxygen, the degradation of benzene, toluene, and xylene by Mn-TiO_2_ can be achieved with the yields 92.6%, 82.9%, and 75%, respectively. The red-shift in the band gap energy upon Mn-doped TiO_2_ was observed, which shows the widening of UV-Vis absorption.		[72]
Cerium (Ce)	Visible light degradation of toluene and o-xylene	Sonochemical synthesis	2.89–3.07	Photodegradation efficiencies of Ce-doped TiO_2_ (CeT) gained higher than undoped TiO_2_. The degradation efficiency by the CeT was consistent with the high desired surface area, pore size, and the particle size of the catalysts. Furthermore, the proper amount of Ce on TiO_2_ has less photoluminescent (PL) emission than undoped TiO_2_ and P25, indicating an increase in electron-hole separation and a reduction in photoexcited charge carrier recombination.		[73]

**Table 4 membranes-12-00345-t004:** TiO_2_-based heterojunction photocatalysts for organic pollutant treatment. Several images in the table are reproduced with permission from Elsevier, 2022.

Material(s)	Pollutant Target(s)	Result(s)			Mechanism	Reference
		SSA (m^2^/g)	Band Gap (eV)	Rate Constant (min^−1^)		
CuO/WO_3_/TiO_2_	Phenol; 4-chlorophenol (4-CP); 3-phenyl-1-propanol (3-P-1-P)	41.1431		0.0621 (phenol); 0.017 (4-CP); 0.0108 (3-P-1-P)	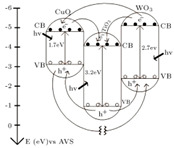	[82]
TiO_2_-WO_3_/ZSM-5	xylene	313.8		0.02826	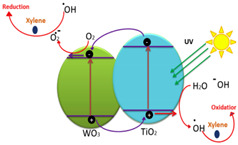	[83]
TiO_2_/CdS	Phenol	146.5		0.01266	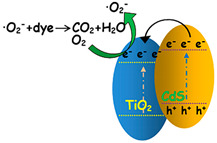	[84]
TiO_2_(A)/TiO_2_(R)/ZnO	4-CP		3.0	0.0371 (UV); 0.0152 (Visible)	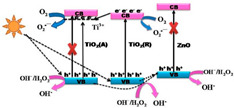	[85]
TiO_2_(A)/TiO_2_(R)/SnO_2_	4-CP		3.0	0.0234 (UV); 0.0102 (Visible)	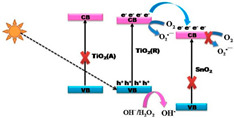	[85]
CeO_2_/TiO_2_	Phenol	76.9	2.88	0.0302	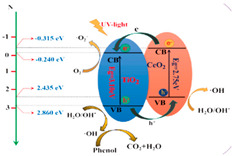	[86]

**Table 5 membranes-12-00345-t005:** Photodegradation reaction involving Fe^3+^-TiO_2_-Zeolite nanocomposite [88].

Processing Step	Reaction
1. Charge carrier separation	Fe^3+^-TiO_2_-Zeolite → Fe^3+^-TiO_2_-Zeolite (e^−^ + h^+^)
2. Trapping and electron transfer	Fe^3+^ + e^−^ → Fe^2+^ Fe^2+^ + O_2_ → Fe^3+^ + O_2_^•^
3. Hydrogen peroxide formation	O_2_^•^ + H^+^ + e^−^ → HO_2_^•^ HO_2_^•^ + Fe^2+^ + H^+^ ⇌ H_2_O_2_ + Fe^3+^
4. Oxidation of Fe^2+^	Fe^2+^ + h^+^ → Fe^3+^ Fe^2+^ + H_2_O_2_ + H^+^ → Fe^3+^ + HO^•^ Fe^2+^ + H_2_O + HO^•^ → Fe^3+^ + H_2_O_2_ H_2_O_2_ + 2e^−^ → 2HO^•^
5. Interface charge transfer	Organic compounds + [Fe^3+^-TiO_2_-Zeolite (HO^•^, O_2_^•^, HO_2_^•^)] → Degraded product

**Table 6 membranes-12-00345-t006:** The specific surface area, average pore size, and pore volume of various photocatalysts [94].

Properties	Samples			
	NB	C-NB	PdO-NB	PdO-C-NB
BET surface area (m^2^/g)	30.82	34.29	42.72	48.67
Pore size (nm)	7.73	8.37	7.95	11.15
Pore volume (m^3^/g)	0.12	0.14	0.17	0.27

Notation: NB = TiO_2_ nanotubes; C-NB = corroded TiO_2_ nanotubes; and PdO-NB = PdO-loaded TiO_2_ nanotubes.

**Table 7 membranes-12-00345-t007:** The recent advance of TiO_2_-based PM.

Photocatalyst	Membrane and Module	Fabrication Method	Result(s)	Reference
TiO_2_/graphene oxide (GO)	Cellulose acetate/cellulose triacetate (CA/CTA) Flat-sheet	Immersion precipitation	- The appearance of TiO_2_/GO enhanced permeate flux and salt rejection of membrane up to 1.81 ± 0.05 L/(m^2^ h bar) (LMH) and 90%, respectively. The higher hydrophilicity of the TiO_2_/GO incorporated membrane had better antifouling performance than the neat membrane, which can be maintained until 80% of the initial flux after 180 min. - Overall, TiO_2_/GO PM revealed high removal of BTEX species, which was better than the conventional membrane.	[97]
TiO_2_	Polyvinylidene fluoride (PVDF) Hollow-fiber	Dry-wet jet spinning	- The addition of TiO_2_ improved several physicochemical properties of the resulting membrane, such as hydrophilicity, porosity, pore area, and tensile strength. - Excess amount of TiO_2_ led to aggregation and closure of membrane pore that could reduce surfactant rejection and flux. - The UV-A illumination of TiO_2_/PVDF PM with 2 wt% of TiO_2_ created a synergistic effect of photodegradation and separation process that possessed the highest surfactant rejection and membrane flux of 66.73% ± 0.76% and 47.95 ± 1.34 L/m^2^h, respectively.	[98]
TiO_2_/carbon nanotubes (CNT)	Commercial PVDF Flat-sheet	Surface coating	- According to the scanning electron microscope (SEM) observation after OPW treatment under UV irradiation for 12 h, TiO_2_/CNT coated membrane was able to reduce the amount of accumulated pollutant with a proven degradation performance due to the photocatalytic ability of TiO_2_/GCN. - Not only photodegradation but also TiO_2_/CNT coated membranes gained 2.5-times higher flux than the pristine membrane, which can maintain high flux recovery ratio until reach the highest oil removal efficiency and membrane flux up to >98% and 362 L m^−2^ h^−1^, respectively.	[99]
TiO_2_	Polyacryllonitrile (PAN) nanofibers Flat-sheet	Electrospray dispersion	- SEM results showed a high uniform dispersion of TiO_2_ nanoparticles on the PAN surface, indicating that electrospray dispersion was effectively used to incorporate TiO_2_ nanoparticles into composite nanofibrous membranes with an average diameter of 900 nm. - The degradation and filtration efficiencies of TiO_2_-dispersed PAN nanofiber membranes improved significantly, reaching around 97.9% conversion of toluene and 97.5%, respectively. These findings were attributed to the high amount of CO_2_ produced, up to 2842.84 mg/m^3^, as well as the high pore distribution at around 1–4 microns.	[100]

**Table 8 membranes-12-00345-t008:** Amount of nanoparticles present in aqueous media for polymeric membranes not exposed and exposed to UV illumination [101].

Membrane	3 h UV	6 h UV	6 h (Dark Controls)
	Particles (10^8^/mL)		
Polyetersulfon (PES)	0.75	2.4	0.2
DK	1.5	2	0.16
BW30-400 (BW)	0.23	2.2	0.39
Cellulose acetate (CA)	0.16	0.12	0.33
NYLON	0.27	0.53	0.13

## Data Availability

Not applicable.

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
