# Peer review of "A Review of Titanium Dioxide (TiO2)-Based Photocatalyst for Oilfield-Produced Water Treatment"

_membranes, 2022, doi:10.3390/membranes12030345_

Round 1

Reviewer 1 Report

The manuscript submitted by the authors has scientific values for the related researchers. Studying of this manuscript shows that the overall framework of the article is well designed, and it is well written. The description and discussion in each title are good and express all aspects of the subject. In general, from my point of view, this work can be accepted for publication in a revised form, after considering the following comments.

Although section 3 (Conclusion and Future perspectives) is one of the most important parts of this manuscript, the authors have failed to give a clear perspective of the future development, as well as a full description of the existed challenges. The readers will be concerned with both a clear perspective and the existing challenges. In fact, they should add a section named “Challenges and future development”.

There are some minor grammatical errors that should be resolved. Authors should carefully check and revise their manuscript before submitting a revision.

Reviewer 2 Report

The presented Manuscript entitled “A review of titanium dioxide (TiO2) based photocatalyst for oilfield produced water treatment” describes the properties of modified TiO2, seeking to analyze how to increase the efficiency in the applied photocatalytic process, in particular in oilfield produced water (OPW).

The article is very complete.

Please insert the band gap values, as for example, in table 3.

Reviewer 3 Report

Comments on manuscript no. membranes-1622631  "A review of titanium dioxide (TiO2) based photocatalyst for oilfield produced water treatment"

Fairly speaking, the manuscript submitted by the authors has a high scientific value for the related researchers, I think. Studying of this manuscript shows that the overall framework of the article is well designed, and it is almost well written. By comparing this manuscript with related work done in this field, it is found that the authors have shun redundant studies.

Thus, I recommend publication of this work after revising the manuscript according the following comments:

  • A suitable graphical abstract should be prepared and included for this work.
  • Some parts of the manuscript need up to date references and more discussion.
  • All parts of the manuscript should be carefully checked from the linguistic point of view. This is the main problem with the current manuscript.
  • In Fig. 6, the numbering of different hvs should be corrected in the legend.

Reviewer 4 Report

see the file

Round 2

Reviewer 3 Report

Accept